# Machine Learning Adoption in Educational Institutions: Role of Internet of Things and Digital Educational Platforms

**Jiuxiang Li [1],\* and Rufeng Wang [2],\***

1   Social Studies Education, Sunchon National University, Sunchon 57922, Jeollanam-do, Republic of Korea
2   College of Physical Education, South-Central Minzu University, Wuhan 430074, China
\*   Correspondence: lijiuxiang17@163.com (J.L.); wangrufeng@mail.scuec.edu.cn (R.W.)

**Abstract:** The ever-increasing development of information technologies has led to the adoption of advanced learning techniques. In this regard, e-learning and machine learning are two of the emerging instructional means for educational institutes. The current study investigates the role of the Internet of Things (IoT) and digital educational platforms (DEPs) in the adoption of machine learning. The present research additionally investigated the function of DEPs as mediators between IoT and machine learning adoption. The department chairs or heads of 310 departments at 91 Chinese institutions provided the information. In order to analyze the data, we used SPSS 25.0 and SEM (structural equation modeling). The results demonstrated how crucial an impact IoT has on DEPs and the uptake of machine learning. DEPs directly affect machine learning adoption and also act as mediators. The findings also support the mediating role of DEPs in the IoT and machine learning adoption link. The current study contributes to both theory and practical management by examining how IoT is helpful for achieving machine learning adoption. Based on the responses of 91 educational departments, this is a unique study of the mechanisms to achieve machine learning adoption through IoT and DEPs.

**Keywords:** internet of things; digital educational platforms; machine learning adoption; educational institutes

## 1. Introduction

Lv [1] stated that in the recent decades, researchers and academics have paid more attention to machine learning as an instructional method. Particularly in the era of digitalization, machine learning is considered an alternative to the conventional learning model that makes use of the internet to deliver education in an unconventional manner and has become a strategic approach for educational institutes (Lallez [2]). Mastan et al. [3] documented that the machine learning approach that facilitates the accumulation of knowledge for each and every person plays a vital role in the accomplishment of educational institutions. Furthermore, Shahbazi and Byun [4] highlighted that machine learning adoption can transform traditional methods of classroom instruction into digitalized instruction methods that can be used to enhance the performance of educational institutions. According to Almaiah et al. [5], massive improvements in digital technologies change the educational environment, models, and practices, and particularly the learning paradigm. Therefore, educational entities that use various undated learning technologies to teach, learn, and interact with potential students are relevant for machine learning adoption, as suggested by Lallez [2].

Lee and Lim [6] stated in their study that advancements in technology exert pressure on educational entities to cope with these changes and adopt machine learning mechanisms for teaching and learning processes. Meanwhile, Hessen et al. [7] highlighted the development of digitalization as instigating traditional ways of learning and teaching to undergo emerging changes. According to Almaiah et al. [5], in the digital educational environment,

IoT enables the integration of academic data, which can be used for the transformation of learning processes and flourishing e-learning activities. Zeadally et al. [8] stated that in the era of digitalization, organizations mostly develop technologies such as the Internet of Things (IoT) in order to improve their capacity for improvising and learning in the future. Furthermore, educational institutes can digitalize their practices and improve e-learning activities by adopting emerging technology such as machine learning. The purpose of the current study is to highlight the role of IoT for the improvement of e-learning.

Ashima et al. [9] documented that although organizations execute their instructional activities using various technological applications, deliberation on the role of IoT has been ignored. In the digital age, the multidirectional mechanisms of advanced technologies enable educational organizations to interact with learners, particularly students, in order to collaborate and participate in learning. Sinha et al. and Huang and Li [10,11] pointed out in their studies that in the recent scenario of a digitalized educational environment, organizations have put efforts into the procedures of IoT and building for the DEPs attached to these multi-directional IoT programs in their strategic educational planning. De Vas et al. [12] highlighted the important of IoT and stated that IoT has gained wider attention as it has been considered a valuable strategic movement for educational entities, which increases the mechanism of digital educational platforms (Huang and Li [11]). Therefore, the focus of the current study is to explore the role of IoT in the improvement of digital educational platforms.

On the other hand, Gillet et al. [13] stated that DEPs improve the students' and learners' attachment to the learning processes of educational institutions, as well as enable the students to accept the changes made by the institutions in their learning processes. Liu and Ardakani [14] documented that when students are highly attached to the institution regarding the execution of digital educational activities, they are more inclined to record higher machine learning adoption. Existing studies such as Almaiah et al. and Huang and Li [5,11] have highlighted that educational institutions with IoT provide massive opportunities for the involvement of students in their strategic learning decisions. According to findings of Sinha and Dhanalakshmi [10], IoT raises DEPs; this ultimately improves machine learning adoption. The current study highlights the mediating role of DEPs in the relationship between IoT and machine learning adoption.

The current study adds to the existing body of knowledge by exploring the direct effect of IoT on the adoption of machine learning. So far, limited studies highlighted the role of IoT for the improvement of machine learning mechanism. As per the suggestions of Zeadally and Tsikerdekis [8], IoT gains importance for the direction of machine learning. However, IoT is critical but not sufficient for the adoption of machine learning directly. Therefore, the current study contributes to the existing body of literature through intervening role of DEPs. A total of five parts make up the present study. The first part serves as an introduction, while the second discusses the context, the literature reviews, and the research methods. Section 3 of this research paper presents and discusses the study's findings. The study's findings and commentary will be presented in Sections 4 and 5, respectively.

## 2. Literature Review

### 2.1. IoT and Machine Learning Adoption

Liu et al. [15] defined IoT as interrelated systems and objects that are connected with each other with the help of the internet and enable the collection and transfer of data. Simply IoT is defined as the interconnection of various digital machines through the internet for the purpose of data accumulation (Gulati et al. [16]). On the other hand, Saleh et al. [17] stated that machine learning adoption is meant as the highest advancement in the learning process, which requires both algorithms and datasets. With the help of IoT mechanisms, required data and information can be easily collected and disseminated, which facilitates the adoption of machine learning. According to Malik et al. [18], students participating in online learning are provided with access to a diverse array of resources online, which increases the likelihood that they may acquire new information. De-Arteaga et al. [19]

documented that the capacity to easily keep up with new information and abilities, for instance, is one of the primary benefits that come with online learning. Machine learning was first identified in the 1950s as an area of AI study. Unfortunately, despite the fact that machine learning has been around since the 1950s, it has made no real progress. This discipline had its comeback in the 1990s, but it has made significant strides since then. As time goes on, this area will only continue to improve. This development is the end result of efforts to analyze and interpret ever-growing data sets. The optimal model can be selected from existing data with the use of fresh information. Machine learning can benefit from this, as per Almaiah et al. [20]. The more data there are, the more research into machine learning will be conducted.

Almaiah et al. [5] suggested that IoT infrastructure allows for the adoption of machine learning within educational institutes with the help of monitoring devices and sensors that imprison objects and the data of people, platforms to process and store data, and communication networks through the internet. Zeadally and Tsikerdekis [8] stated that machine learning adoption is dependent on IoT sharing the necessary information and data via the internet. Huang and Li [11] highlighted that IoT-enabled processing and data collection take place via digital applications that rely on the internet for connectivity. Gillet et al. [13] also documented that IoT used the various digital applications to improve the data set required for the adoption of machine learning as IoT devices store, analyze, and collect huge amounts of data. When data are attained with the help of IoT devices, adoption of machine learning becomes feasible for educational entities. According to Almaiah et al. [20], IoT is a connection of physical objects that collect, store, and share data through various technologies and techniques. IoT provides advanced options for enhancing the use of data and is concerned with machines and computers that enable the adoption of machine learning.

**Hypothesis 1 (H1).** *IoT has a positive relationship with machine learning adoption.*

### 2.2. IoT and DEPs

According to Sinha and Dhanalakshmi [10], DEPs refer to the network made up of students, teachers, and different processes that enable educational entities to quickly produce and activate digital services for the sake of learning. As per the findings of Huang and Li [11], IoT facilitates the connecting of all stakeholders within an organization through various platforms, such as social network services. Liu et al. [15] highlighted that emerging technologies enable the sharing of information among the members of a specific network with the help of connected devices. Chen et al. [21] documented that IoT not only involves the connecting of various technologies for the collecting and disseminating of valuable information but also offers opportunities for the development of DEPs among the institutional members. The same idea is depicted by Zeadally and Tsikerdekis and Almaiah et al. [8,20], that IoT provides a foundation for the development of DEPs within educational institutions. According to Akhter and Sofi [22], IoT provides the basis for connections among people that can easily share valuable information and create DEPs. De-Arteaga et al. [19] stated that the growth of the IoT allows for connections to be made using digital devices for data transmission, which improves online platforms. Huang et al. and Liu and Ardakani [11,14] suggested that IoT speeds up the process of student and teacher participation in digital platforms through the use of emerging digital technologies. Therefore, our study disputes that digital platforms are largely based on the sound mechanisms of organizational IoT.

**Hypothesis 2 (H2).** *IoT has a positive relationship with DEPs.*

### 2.3. DEPs and Machine Learning Adoption

Huang and Li [11] stated that advanced development in the field of information technology has led to a rise in digital educational resources. On the other hand, Liu and

Ardakani [14] documented that machine learning is one of the approaches being adopted by various educational institutions in order to cope with these changes. Therefore, adaptive e-learning and recommendation algorithms in customization systems are responding to these emerging changes in a significant way. Machine learning requires input data, algorithms, evaluations, and results [23]. On the other hand, all the concerned parties of educational institutions that are connected with each other and communicate information through online platforms are recognized as DEPs.

Malik et al. [18] documented that with successful adoption of machine learning, there is a need for data and information, which can be easily accumulated through the existing DEPs. DEPs facilitate the adoption of machine learning for required input data and information from various entities linked by digital platforms. Li [24] stated that DEPs facilitate communicating and sharing information among institutional members such as students, teachers, and administrative staff. Furthermore, Almaiah et al. [23] suggested that DEPs bring together teachers and students and provide the foundation for an e-learning process that facilitates the adoption of machine learning. Malik et al. and Li [18,24] highlighted that these educational platforms sustain the digital transformation of educational institutions and have adopted various technological tools to reduce the barriers to the adoption of machine learning.

IoT is developing because of the proliferation of digital platforms for the dissemination of information and the generation of novel ideas, as stated by Chen et al. [21]. Waheed et al. [25] stated that machine learning adoption requires advancements in learning processes that depend on sound DEPs among the members of educational institutions. Machine learning adoption carries benefits for both teachers and students at educational institutions for effective learning (Malik et al. [18]). However, IoT provides a foundation for the development of DEPs among the members of educational institutions through network connectivity, which in turn plays a role in the adoption of machine learning. DEPs act as a mediator between IoT and machine learning adoption.

**Hypothesis 3 (H3).** *Digital educational platforms are positively associated with machine learning adoption.*

**Hypothesis 4 (H4).** *Digital educational platforms mediate between the Internet of Things and machine learning adoption.*

*2.4. Theoretical Framework*

The aim of all kinds of research study is significant addition to the existing body of knowledge. Contributions of any research to the existing body of knowledge become trustworthy, relevant, and scientifically significant when research study is based on theories. Osanloo and Grant [26] documented that theoretical framework offer study plans mostly based on existing theories adopt by the researchers. Adom et al. [27] stated the importance of theoretical framework for scientific research, as it offers the explanation of related research phenomena.

Theoretical framework of the research study is important for the explanation of relationships among study constructs. Adom et al. [27] suggested that theoretical framework is useful to define the scope of research study. Existing studies in the field of information technologies and particularly for the adoption of new technology mostly based on technology acceptance (TAM) model, TOE (technology, organizational, and environmental) framework, diffusion of innovation (DOI) model, and unified theory of acceptance and use of technology (UTAUT).

The current study was based on UTAUT assumptions for explaining the determinants of machine learning adoption. Venkatesh et al. [28] explained the importance of UTAUT in the field of information technology. Furthermore, Venkatesh et al. [28] documented that UTAUT was based on four components, i.e., facilitating conditions, effort expectancy, performance expectancy, and social influence. In line with the arguments of Venkatesh

et al. [28], in the current study, we formulated the determinants of machine learning adoption based on facilitating conditions one of the components of UTAUT. Facilitating conditions of UTAUT affect use behavior of the individual instead of behavioral intention [29]. The adoption of technology, i.e., machine learning adoption within educational institutions, is largely based on the infrastructure developed by these institutes. In the current study, facilitating conditions can be taken as a determinant of adoption because facilitating infrastructure in the form of IoT provides a foundation for the adoption of machine learning. Furthermore, the social influence factor of UTAUT is concerned with an individual's feelings about others' beliefs about them using the new system [28]. Therefore, social influence can be taken as a factor for technology adoption because social influence helps in maintaining physical distance through digital platforms. In the current study, two determinants of machine learning adoption, i.e., IoT and digital educational platforms, are used. These two determinants are based on the two components, including facilitating infrastructure and social influence, of UTAUT.

Based on the above discussion, The connection between the constructs employed in this analysis is shown in Figure 1. In this research, we examined the relationships between the Internet of Things (as an independent variable), digital educational platforms (as a mediator), and the application of machine learning (as a dependent variable) (dependent variable).

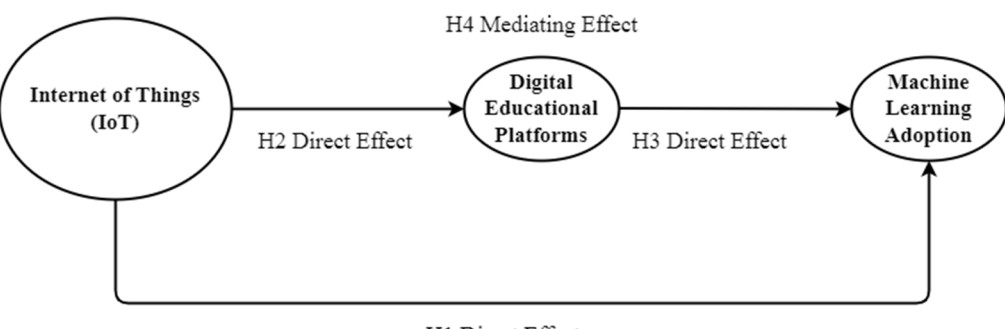

**Figure 1.** Theoretical Framework.

### 3. Methodology

This was a cross-sectional study and questionnaire (See Appendix A) was the main tool used for data collection, which was kept as concise as possible with simple/easy-to-understand wording. Cross-sectional studies can be generalized to the whole population from the basis of a small representative sample. It is a reliable and valid method with which to make conclusions regarding the issue being addressed in the study, paving the way for others to use it as a reference for future studies.

Random sampling technique was used for the selection of sample. The population consists of 91 public-sector universities. For the purpose of study, information about universities was taken from the Ministry of Education of China. In the next step, we approached the universities for the information about various departments. Out of 1678 departments, we selected 310 having IT infrastructure, students' digital platforms, and e-learning infrastructure; most of these departments included information technology, management sciences, and public administration. Table 1 depicts the demographic information of selected departments.

The chairman or head of department was surveyed using an online form built with Google Docs's tools. Online survey tools were developed for this specific reason. We split the survey administration into two waves, T1 and T2, separated by a two-week time frame to avoid the potential for bias caused by the more conventional sampling approach. Selected respondents were sent a link to the online survey platform. For the purpose of facilitating a deeper comprehension of the concepts, the questionnaire was created in both English and Chinese. Between January 2022 and April 2022, email surveys were sent

out to potential participants. T1 (the first wave) consisted of a survey being sent out to a random sample of 310 people. Information on the IoT and DEPs was gathered at TI. Only 312 valid replies were obtained in the first round. The second wave of data collection (T2) began after an interval of two weeks. Only 312 people participated in the second round of the survey. Department chairs were asked to judge how well their teams had embraced machine learning. At long last, we have 271 finished, fully-complete replies.

**Table 1.** Departments characteristics.

|  |  | N | % |
|---|---|---|---|
| | 1–5 | 43 | 15.86 |
| | 6–10 | 109 | 40.22 |
| Department age (in years) | 11–15 | 41 | 15.12 |
| | 15–20 | 44 | 16.23 |
| | Above 20 | 34 | 12.54 |
| | 10–20 | 7 | 2.58 |
| Department size (no. of teaching staff) | 21–25 | 17 | 6.27 |
| | 26–30 | 143 | 52.76 |
| | More than 30 | 104 | 38.37 |
| | Information Technology | 143 | 9.96 |
| Department type (nature) | Management Sciences | 54 | 19.92 |
| | Public Administration | 27 | 52.76 |
| | Others | 47 | 17.34 |

*Measures*

First section of the questionnaire was regarding demographics of the departments which included departments age, type, and size. Details are given in Table 1 below.

The second part of the survey asked about experiences with the Internet of Things, data exchange protocol, and machine learning. There was a five-point Likert scale used for all the questions, with 1 indicating a strong disapproval, 2 a moderate one, 3 a neutral one, 4 an agreement, and 5 a strong agreement. To gather information from the designated university divisions, we used the following scales, which we either modified from existing instruments or created from scratch.

De Vass, et al. [17] created a 10-item scale to assess IoT as an independent variable, and it was employed in this study. Moreover, mediating variable (DEPs) was measured with 8-item scale formulated by Rai and Tang [30]. A 3-item-based construct of machine learning adoption was adapted from Jadhav [31]. Table 2 shows details of constructs.

**Table 2.** Measurement scale.

| Constructs | Scale | Author | No. of Items |
|---|---|---|---|
| Internet of Things | Adapted | De Vass et al. [17] | 10 items |
| Digital Educational Platforms | Adapted | Rai and Tang [30] | 8 items |
| Machine Learning Adoption | Adapted | Jadhav [31] | 3 items |

## 4. Results

In the first phase of data analysis, we established the validity of the constructs. The statistics of dependability are shown in Table 3. Table 3 displays findings that are all above the cutoff value. The results shown in Table 3 validated the validity and reliability of the constructs as the values of Alpha, Loading, CR, and AVE all met the threshold level, showing that there is no problem with reliability or validity.

**Table 3.** Reliability measures.

| Variables | FL | AVE | CR | Cronbach's $\alpha$ |
|---|---|---|---|---|
| Internet of Things | 0.62–0.87 | 0.699 | 0.971 | 0.801 |
| Digital Educational Platforms | 0.67–0.89 | 0.722 | 0.957 | 0.842 |
| Machine Learning Adoption | 0.71–0.91 | 0.786 | 0.973 | 0.884 |

*4.1. Correlation*

The results from the Pearson correlation analysis were then used to establish the degree of association between the various constructs. IBM SPSS Statistics, version 25.0, was used to determine the degree of association between the various components of the research. Correlational statistics were used to verify the link between the variables. Regardless of the statistical significance of the two variables or the degree to which one relies on the other, the direction of the link between them was established by the coefficient of correlation. All the research constructs were positively related, as shown by the correlation coefficients.

Table 4 contains the findings of correlation analysis. Findings shows that IoT has positive association with DEPs (0.29 **) and machine learning adoption (0.19 **). Furthermore, the results of correlation confirmed the positive relationship of DEPs with machine learning adoption (0.33 **).

**Table 4.** Correlations.

| Variables | 1 | 2 | 3 | 4 |
|---|---|---|---|---|
| Internet of Things | 0.84 | | | |
| Digital Educational Platforms | 0.29 ** | 0.81 | | |
| Machine Learning Adoption | 0.19 ** | 0.33 ** | 0.83 | |

** $p < 0.01$, two-tailed.

*4.2. Hypotheses Testing*

The next phase was using AMOS 7.0 software to determine the relevance and intensity of dependence between variables after the validation of the direction and linkages among research components. By using a structural equation modeling (SEM) strategy, we used AMOS 7.0 to evaluate hypotheses in this work. Data results are shown in Table 5.

**Table 5.** Result of path analysis.

| Paths | Estimates | SE | C.R |
|---|---|---|---|
| Machine learning adoption ← IoT | 0.17 | 0.063 | 2.698 ** |
| DEPs ← IoT | 0.29 | 0.057 | 5.087 ** |
| Machine learning adoption ← DEPs | 0.33 | 0.059 | 5.593 ** |

Note: IoT (Internet of Things); DEPs (digital educational platforms); SD (standard deviation); ** $p < 0.01$, two-tailed.

The results depicted in Table 5 entail that IoT is related to machine learning adoption ($\beta = 0.17$, t = 2.698, $p < 0.00$). Based on these findings, study H1 was accepted. Furthermore, the findings also confirmed the effect of IoT on DEPs ($\beta = 0.29$, t = 5.087, $p < 0.00$). Based on these findings, study H2 was accepted. Moreover, the findings revealed that DEPs are significantly and positively associated to machine learning adoption ($\beta = 0.33$, t = 3.593, $p < 0.00$). Based on these findings, study H3 was accepted.

To check out the indirect effects of IoT on machine learning adoption, we applied the normal test theory approach. This approach provides the statistics of direct, indirect, and total effects. Outcomes of the normal test theory approach are presented in Table 6. Table 6's results elucidated the direct impact (0.32 **) and indirect effect (0.32 0.17 = 0.15) of IoT on machine learning adoption through DEPs. Sobel test results (Z = 3.72) confirmed the intermediate impact of IoT on the diffusion of DEP-based machine learning. These results convince us that Study H4 is valid.

**Table 6.** Outcomes of indirect effect of DEPs.

| Mediation Models | Total Effect | | | Direct Effect | | | Indirect Effect | | |
|---|---|---|---|---|---|---|---|---|---|
| | | | | | | | Normal Test Theory | | |
| | *B* | *T* | *p* | *B* | *T* | *p* | *B* | *Z* | *p* |
| IoT → DEPs → Machine learning adoption | 0.32 | 8.72 | 0.00 | 0.17 | 1.39 | 0.09 | 0.15 | 3.72 | 0.00 |

Note: IoT (Internet of Things); DEPs (digital educational platforms).

## 5. Discussion

The aim of the current study was to identify how IoT affects DEPs and to what extent DEPs help the adoption of machine learning in educational institutes. Our study consisted of four hypotheses, which explained the association between IoT, DEPs, and machine learning adoption. Study H1 explained the direct effect of IoT on the adoption of machine learning (0.17 **). The findings revealed that IoT is avaluable means of increasing opportunities for educational institutions to engage them and adopt machine learning. Arevalo-Lorido et al. [32] suggested that technological development continues to rapidly grow, which changes the learning mechanisms of educational institutions, including machine learning adoption. IoT helps educational institutions in giving information which is computer-generated to all the stakeholders.

Study H2 shows the direct relationship between IoT and DEPs (0.29 **). The findings suggested that IoT provides opportunities to the organizations for the development of digital platforms. Previous findings, e.g., by Makkar and Kumar [33], support that IoT is a significant predictor of e-learning activities of educational institutions. Waheed et al. [25] documented that IoT facilitates organizations in building digital platforms and adopting machine learning. The prior findings of Nykyri et al. [34] suggested that IoT is a set of all processes and applications of technological resources to develop platforms for acquisition and dissemination of data and information.

The third hypothesis links DEPs and machine learning adoption. DEPs act as important means for the adoption of machine learning. This study finding is constant with previous research, e.g., by Huang and Li [11], about digital platforms through which organizations can easily adopt e-learning and machine learning. Fourth, DEPs mediate between IoT and the adoption of machine learning. Digital platforms generate the required information by using IoT and provide a foundation for the adoption of machine learning. It is impossible for machine learning to function without its fundamental building blocks, which are comprised of data sets, algorithms, evaluation, and output. Liu et al. [14] documented that educational institutions try to build up DEPs by enabling IoT, which is apleasing and desirable factor for the adoption of machine learning. These findings revealed that DEPs in response to IoT positively influence the adoption of machine learning.

### 5.1. Theoretical Contribution

The work at hand considerably and conceptually adds to the current literature of information technology and e-learning. This work also considerably adds to the current body of information by increasing the UTAUT. The main contribution is the formulation of a model based on the facilitating conditions and social influence assumptions of UTAUT, which tested the IoT as a determinant of DEPs and machine learning adoption. There are limited research studies that consider the technological facilitating factors based on UTAUT for boosting machine learning adoption. Limited studies focused on the assumptions of UTAUT to highlight the determining factors for the adoption of machine learning. Furthermore, a comprehensive research model based on the assumptions of UTAUT was developed for educational institutes to test both the direct and indirect impact of IoT on machine learning adoption.

Furthermore, the current study also adds to the existing body of knowledge by explaining the role of IoT in building DEPs, which in turn enhances the e-learning approach

of educational institutions. Huang and Li [11] document that DEPs are an important mechanism and important for the thriving formulation of a new model for e-learning. Existing studies in the relevant field ignore the role of DEPs with respect to their determinants and outcomes. As a result, the current study took this into account, filled this research gap, and focused on IoT as a potential determinant of DEPs.

### 5.2. Practical Implications

The findings of the current study provide valuable implications for practice. First, the findings suggest that educational institutes can adopt the machine-learning method with the help of the IoT and through DEPs. When these institutions respond positively to digitization, adoption of machine learning is possible. Second, according to this study, educational institutions give attention towards the facilitating infrastructure, such as IoT, to make it possible to build digital platforms for the exchange of required information. The findings suggested that the managements of educational institution stake serious action for the development of infrastructure that supports the adoption of e-learning and machine learning procedures. Finally, the findings of the current study also suggested for the managements in practice to build educational digital platforms for the smooth functioning of learning activities. Such platforms provide a foundation for the exchange of valuable information required for the formulation and implementation of innovative e-learning techniques. Therefore, the managements can easily promote and adopt machine learning mechanisms in their institutes through proper IoT and DEPs.

### 5.3. Limitation and Future Recommendations

There are limitations to the current study that might be suggestions for future studies. Before moving on, it should be noted that the present study focused only on Chinese universities. However, in order to generalize the study's conclusions, future research may expand its scope to include universities in other developed countries. Second, although we used a cross-sectional approach to gather data and evaluate our findings, qualitative studies may help us uncover other elements that might help push the needle on machine learning's widespread adoption. Third, this study used UTAUT to examine the influence of IoT on DEPs and their adoption of ML; future research might benefit from exploring the role of alternative theories, such as a DOI and the TAM model.

## 6. Conclusions

According to the current study, IoT isone of the important technology-centric mechanisms that allow members of educational institutions to easily interact with one another. IoT is one of the important mechanisms that ensure interaction and collaboration among teachers and students. The most constructive aspect of IoT is DEPs. The findings revealed positive effects of IoT on both DEPs and the adoption of machine learning. Furthermore, findings also suggested that DEPs also positively affect the adoption of machine learning. In summary, we concluded that IoT mechanisms increase DEPs, which become the reason for the building of connections among members for the collection and sharing of the required data and information necessary for the adoption of machine learning.

**Author Contributions:** Both authors (J.L. and R.W.) contributes equally. All authors have read and agreed to the published version of the manuscript.

**Funding:** This research received no external funding.

**Institutional Review Board Statement:** The study was conducted in accordance with the Declaration of Helsinki and approved by the Institutional Review Board (or Ethics Committee) of Sunchon National University RN-34/98-2022 dated: 13 August 2022.

**Informed Consent Statement:** Informed consent was obtained from all subjects involved in the study.

**Data Availability Statement:** Data is available on request.

**Conflicts of Interest:** The authors declare no conflict of interest.

## Appendix A. Measurement Items

Internet of Things

1. Our firms may provide item-level identifiers made possible by the Internet of Things.
2. Using the Internet of Things, our company is able to give unit-level identifiers (such as containers, boxes, and pallets).
3. Our company has the ability to automatically acquire data for the purposes of tracking and tracing employees and other operational elements.
4. We have the ability to gauge the circumstances under which businesses operate, their operations, and their activities.
5. The Internet of Things (IoT) allows for the remote management of commercial operations. IoT enables us to provide real-time information to optimize business activities.
6. As a result of the Internet of Things, we are able to provide real-time insight into corporate processes.
7. We have the ability to provide significant amounts and types of data for the use of data analytics in operational and strategic planning.
8. The Internet of Things allows for better inter-operator communication and collaboration.
9. Our company is able to provide item-level identification because of the Internet of Things.

Digital educational platforms

1. Information from our educators is readily available on our site.
2. Our platform provides seamless connection between our students and teachers.
3. Our firm can auto-captures data to monitor, track and trace operational activities and people.
4. Our technology makes it simple to compile data from classroom instructors and students.
5. Our system can quickly be modified to accommodate additional users, such as instructors and students.
6. IT applications or features may be quickly integrated onto our platform.
7. Our system adheres to standards recognized by the vast majority of our present and future collaborators.
8. Our system is built from re-usable software modules that may be integrated into a wide variety of academic programs.

Adoption of machine Learning

1. The majority of programs at my school are planning to use some kind of machine learning.
2. The likelihood of future machine learning adoption by departments at my institution is high.
3. What do I think the timeline is for the departments in my academic division to implement machine learning?

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
