# Peer review of "Machine Learning Adoption in Educational Institutions: Role of Internet of Things and Digital Educational Platforms"

_sustainability, doi:10.3390/su15054000_

Round 1
Reviewer 1 Report
Introduction:
What is the objectives and purpose of this research? research gap and objectives should be better presented and explained at the end of introduction section. The introduction section could be improved also by adding more related and recent reference regarding the subject of interest.
Literature Review and theoretical framework section:
I did not see literature Review and theoretical framework section, there is missing a strong section on review of recent relevant literature, please consider related works and what are the main differences between this study and previous. In addition, the study framework encompassed one determinant that influenced the DEPs and machine learning adoption. The way of extracting these factors should be mentioned. Kindly explain the reason for the addition of the factors. What was your theoretical base? need more clarification as well.
Methodology
The methodology seems underdeveloped. The research methodology is not fully described. A more detailed description and explanation is required. I suggest reviewing the method.
In Table 1. Respondents’ Characteristics, but in the table all statistics related to department (who is the respondent?).
In the paragraph before Table 2. Measurement Scale page 6 line 194, authors claimed (A 03 items based construct of of machine learning adoption was adapted from Jadhav [27]). However, referring to Table 2. the number of items are 4.
The questionnaire should be listed in Appendix. It is the most important data source in this study. I have to see the items to tell if the research is well designed.
Results and hypotheses testing:
Data analysis technique need to be expanded and further explanation to be added. How analysis was done? readers want to see more research process, how did you get the results.
Discussion:
Discussion should be strengthened. Explain clearly what the result of the research is and how it differs from previous research.
Conclusion:
Theoretical contribution and practical implications need to be more improved. This manuscript lacks of clear Implications. I suggest elaborating much more on the theoretical and practical (managerial) implications and their impacts.
limitations of the research and future research directions.
The limitations and the recommendations of future research in the area also need to addressed. It is worthy to create new section/s to include the limitations and the recommendations of future research.
Some references you might find them useful:
1- Examining the Impact of Artificial Intelligence and Social and Computer Anxiety in E-Learning Settings: Students’ Perceptions at the University Level.
2- Performance Investigation of Principal Component Analysis for Intrusion Detection System Using Different Support Vector Machine Kernels.
3- Measuring institutions’ adoption of artificial intelligence applications in online learning environments: integrating the innovation diffusion theory with technology adoption rate.
Finally, the number of references are small. Therefore, after considering my aforementioned comments, the number will be increased by default.
Best of luck
Author Response
|
Reviewer 2 Comments |
Changes incorporated by Authors
|
|
I would strongly advise the authors to reconsider the use of "methods" three consecutive times in three lines (12-13-14) The reviewer is sure that just using the digits to refer the reader to the source without mentioning the source itself (e.g., the name of the scholar) is a misleading and sometimes frustrating option: why should the reader turn the pages each time he/she meets a source? From line 21 to line 41 (13 lines) we came across six sources. Do we need to turn to the end of the article each time to see who it was? We suggest saying "Lv, C. et.al [1] states that...." looking at (reading) lines 42-51 (and further), we see sentences and references to their sources and all these look like mottoes. How are they different or close? Are they contradicting another author's view(s) or adding something? There are typos, such as lack of a blank after the bracket. (line 30, 56 and further). line 102 ...internet.IoT-enabled... (possibly a blank is needed)
lines 174 -177 are repeating each other telling about 310 questionnairs line 194 states that "A 03 items based consturctof of machine learning adoption was adapted from Jadhav [27]. At the same time, Table 2 states 4 items. Could the athors carry out the arithmetics correctly? line 226 "stratistics" is a new word line 227 "nornal test" norNal is a new word line 229 ... effect (0.32 - 0.17 = 0.15), of IoT on machine learning... (extra comma) line 234 The contribution of our study has try to identify (grammar to be checked: perfect simple)line 241 IoT is complete process of maintaining interconnected system (consider using an article) line 244 Second hypothesis support that IoT help in development of digital platforms though which require information are share and disseminate among institutional members including students. (We apologize, however this sentence is in wrong English). Strictly speaking? Discussion is a set of English words, having very little sense, or being only citations of some sources. |
These changes are incorporated by rewriting the abstract.
These changes are incorporated.
These changes are incorporated
These changes are incorporated.
These changes are incorporated by rewriting the paragraphic
This typing mistake is corrected.
These mistakes are corrected the manuscript
These changes are incorporated by rewriting the sentences
Changes are incorporated
Changes are incorporated in the discussion section |

Reviewer 2 Report
I would strongly advise the authors to reconsider the use of "methods" three consecutive times in three lines (12-13-14)
The reviewer is sure that just using the digits to refer the reader to the source without mentioning the source itself (e.g., the name of the scholar) is a misleading and sometimes frustrating option: why should the reader turn the pages each time he/she meets a source? From line 21 to line 41 (13 lines) we came across six sources. Do we need to turn to the end of the article each time to see who it was? We suggest saying "Lv, C. et.al [1] states that...."
looking at (reading) lines 42-51 (and further), we see sentences and references to their sources and all these look like mottoes. How are they different or close? Are they contradicting another author's view(s) or adding something?
There are typos, such as lack of a blank after the bracket. (line 30, 56 and further).
line 102 ...internet.IoT-enabled... (possibly a blank is needed)
lines 174 -177 are repeating each other telling about 310 questionnairs
line 194 states that "A 03 items based consturctof of machine learning adoption was adapted from Jadhav [27]. At the same time, Table 2 states 4 items. Could the athors carry out the arithmetics correctly?
line 226 "stratistics" is a new word
line 227 "nornal test" norNal is a new word
line 229 ... effect (0.32 - 0.17 = 0.15), of IoT on machine learning... (extra comma)
line 234 The contribution of our study has try to identify (grammar to be checked: perfect simple)
line 241 IoT is complete process of maintaining interconnected system (consider using an article)
line 244 Second hypothesis support that IoT help in development of digital platforms though which require information are share and dissiminate among institutional members including students. (We apologize, however this sentence is in wrong English).
Strictly speaking? Discussion is a set of English words, having very little sense, or being only citations of some sources.
line 275 digitalizatio literature (new word or in Italian/Spanish?
Author Response
Dear Editor
Manuscript ID: sustainability-2186364
The paper titled ‘Machine Learning Adoption in Educational Institutions: Role of Internet of Things and Digital Educational Platforms’ has been reviewed changes are incorporated and highlighted in the revised manuscript:
|
Reviewer 2 Comments |
Changes incorporated by Authors
|
|
I would strongly advise the authors to reconsider the use of "methods" three consecutive times in three lines (12-13-14) The reviewer is sure that just using the digits to refer the reader to the source without mentioning the source itself (e.g., the name of the scholar) is a misleading and sometimes frustrating option: why should the reader turn the pages each time he/she meets a source? From line 21 to line 41 (13 lines) we came across six sources. Do we need to turn to the end of the article each time to see who it was? We suggest saying "Lv, C. et.al [1] states that...." looking at (reading) lines 42-51 (and further), we see sentences and references to their sources and all these look like mottoes. How are they different or close? Are they contradicting another author's view(s) or adding something? There are typos, such as lack of a blank after the bracket. (line 30, 56 and further). line 102 ...internet.IoT-enabled... (possibly a blank is needed)
lines 174 -177 are repeating each other telling about 310 questionnairs line 194 states that "A 03 items based consturctof of machine learning adoption was adapted from Jadhav [27]. At the same time, Table 2 states 4 items. Could the athors carry out the arithmetics correctly? line 226 "stratistics" is a new word line 227 "nornal test" norNal is a new word line 229 ... effect (0.32 - 0.17 = 0.15), of IoT on machine learning... (extra comma) line 234 The contribution of our study has try to identify (grammar to be checked: perfect simple)line 241 IoT is complete process of maintaining interconnected system (consider using an article) line 244 Second hypothesis support that IoT help in development of digital platforms though which require information are share and disseminate among institutional members including students. (We apologize, however this sentence is in wrong English). Strictly speaking? Discussion is a set of English words, having very little sense, or being only citations of some sources. |
These changes are incorporated by rewriting the abstract.
These changes are incorporated.
These changes are incorporated
These changes are incorporated.
These changes are incorporated by rewriting the paragraphic
This typing mistake is corrected.
These mistakes are corrected the manuscript
These changes are incorporated by rewriting the sentences
Changes are incorporated
Changes are incorporated in the discussion section |

Round 2
Reviewer 1 Report
The authors didn’t response to most of my comments, but it was only highlighting the same text and statements in yellow color without any amendments. Therefore, I have listed below the main point that still need to more improvements:
Literature Review and theoretical framework section:
I did not see literature Review and theoretical framework section, there is missing a strong section on review of recent relevant literature, please consider related works and what are the main differences between this study and previous. In addition, the study framework encompassed one determinant that influenced the DEPs and machine learning adoption. The way of extracting these factors should be mentioned. Kindly explain the reason for the addition of the factors. What was your theoretical base? need more clarification as well.
2.4. Theoretical framework
In this new created section, kindly explain in detail what was your theory basis and why you chose it as the basis of your framework. What are the advantages of adopting it compared to others? Your reasons must be based on valid scientific references.
Methodology
The methodology still underdeveloped. The research methodology still not fully described. A more detailed description and explanation is required. Kindly determine how these number computed and where did you get them (A total of 310 departments of 91 universities were selected on the basis of simple random sampling. Total population was 1678 departments whereas sample size was selected to be 310 following the population/ public universities.).
In general, the methods section is too short: more information on study context and population are needed, study design, inclusion and exclusion criteria, statistical analysis are needed.
In the paragraph before Table 2. Measurement Scale page 6 line 211, authors claimed (IoT used as an independent variable and measure with 9 items scale formulated 211and validated by De Vass et al. [17].). However, referring to Appendix. the number of items are 10. Kindly double check about it.
Results and hypotheses testing:
Data analysis technique still need to be expanded and further explanation to be added as I didnt see any improvements or clarification only the yellow color was highlighted for former mentioned statements. How analysis was done? readers want to see more research process, how did you get the results. You mentioned in 4.2. Hypotheses Testing the following statement (For the purpose of testing study hypotheses we applied Structural equation modeling (SEM) approach. The outcomes of SEM ..........), what do you mean by SEM as there are many techniques under SEM, kindly specify which one of them you have applied in your study.
Theoretical contribution and practical implications need to be more improved as I didnt touch any modification or revising only the yellow color and the subsection number, you only shifted conclusion section to be the last part, in my opinion, this is not improvement or enhancement to the presentation of the paragraph related to the subsections in demand. This manuscript still lacks of clear Implications. I suggest elaborating much more on the theoretical and practical (managerial) implications and their impacts.
Additionally, kindly create new section and name it (limitations and recommendations for future research) and discuss limitations of the study, and your suggestions for further studies in this area of research. It is necessary to add a clear and focused future research to be able to complete further research.
Good luck
Author Response
Manuscript ID: sustainability-2186364
The paper titled ‘Machine Learning Adoption in Educational Institutions: Role of Internet of Things and Digital Educational Platforms’ has been reviewed changes are incorporated and highlighted in the revised manuscript:
|
Reviewer 1 Comments |
Changes incorporated by Authors
|
|
Literature Review and theoretical framework section: I did not see literature Review and theoretical framework section, there is missing a strong section on review of recent relevant literature, please consider related works and what are the main differences between this study and previous. In addition, the study framework encompassed one determinant that influenced the DEPs and machine learning adoption. The way of extracting these factors should be mentioned. Kindly explain the reason for the addition of the factors. What was your theoretical base? need more clarification as well. Theoretical framework In this new created section, kindly explain in detail what was your theory basis and why you chose it as the basis of your framework. What are the advantages of adopting it compared to others? Your reasons must be based on valid scientific references. Methodology The methodology still underdeveloped. The research methodology still not fully described. A more detailed description and explanation is required. Kindly determine how these number computed and where did you get them (A total of 310 departments of 91 universities were selected on the basis of simple random sampling. Total population was 1678 departments whereas sample size was selected to be 310 following the population/ public universities.).
In the paragraph before Table 2. Measurement Scale page 6 line 211, authors claimed (IoT used as an independent variable and measure with 9 items scale formulated 211and validated by De Vass et al. [17].). However, referring to Appendix. the number of items are 10. Kindly double check about it. Results and hypotheses testing: Data analysis technique still need to be expanded and further explanation to be added as I didnt see any improvements or clarification only the yellow color was highlighted for former mentioned statements. How analysis was done? readers want to see more research process, how did you get the results. You mentioned in 4.2. Hypotheses Testing the following statement (For the purpose of testing study hypotheses we applied Structural equation modeling (SEM) approach. The outcomes of SEM ..........), what do you mean by SEM as there are many techniques under SEM, kindly specify which one of them you have applied in your study. Theoretical contribution and practical implications need to be more improved as I didnt touch any modification or revising only the yellow color and the subsection number, you only shifted conclusion section to be the last part, in my opinion, this is not improvement or enhancement to the presentation of the paragraph related to the subsections in demand. This manuscript still lacks of clear Implications. I suggest elaborating much more on the theoretical and practical (managerial) implications and their impacts.
Additionally, kindly create new section and name it (limitations and recommendations for future research) and discuss limitations of the study, and your suggestions for further studies in this area of research. It is necessary to add a clear and focused future research to be able to complete further research. |
|

Reviewer 2 Report
Dear authors
You have done a lot to make the manuscript better. Unfortunately, you did not pay attention to my remark, concerning the entire text, not only lines 21-31.
One of my remarks said:
The reviewer is sure that just using the digits to refer the reader to the source without mentioning the source itself (e.g., the name of the scholar) is a misleading and sometimes frustrating option: why should the reader turn the pages each time he/she meets a source? From line 21 to line 41 (13 lines) we came across six sources. Do we need to turn to the end of the article each time to see who it was? We suggest saying "Lv, C. et.al [1] states that...." looking at (reading) lines 42-51 (and further), we see sentences and references to their sources and all these look like mottoes. How are they different or close? Are they contradicting another author's view(s) or adding something?
Your answer states:
"These changes are incorporated."
We continue reading and see (just a few examples given here):
"In recent decades, researchers and academics have paid more attention to machine learning as an instructional method [1]. (Who is this number 1?) Particularly in the era of digitalization; machine learning is considered an alternative to the conventional learning model that makes use of the internet to deliver education in an unconventional manner and has become a strategic approach for educational institutes [2]. (Who is this number 2?) A machine learning approach that facilitates the accumulation of knowledge for each and every person plays a vital role in the accomplishment of educational institutions [3]. (Who is this number 3?)
In my remark I suggested using the name(s) of the scholars. E.g., Lv, C. et al state that in recent decades, researchers and academics have paid more attention to machine learning as an instructional method [1], while Lallez, R. [2] considers that particularly in the era of digitalization, machine learning is considered an alternative to the conventional learning model that makes use of the internet to deliver education in an unconventional manner and has become a strategic approach for educational institutes. As for Mastan, I. A. et al. [3], the idea is that a machine learning approach that facilitates the accumulation of knowledge for each and every person plays a vital role in the accomplishment of educational institutions.
The entire text needs to be corrected this way. If you have more than one source, then you use one of the scholar's name and then add, that the same idea is depicted by, e.g., A.b, C.D and X.Y [4, 12, 19].
The manner used in the work, shows only manifestations or mottos (every other sentence comes from a source without using the scholar). This way the manuscript resembles a dictionary or an encyclopedia.
This is the only thing that deters me from accepting. However, my remark concerns the whole text.
Author Response
Manuscript ID: sustainability-2186364
The paper titled ‘Machine Learning Adoption in Educational Institutions: Role of Internet of Things and Digital Educational Platforms’ has been reviewed changes are incorporated and highlighted in the revised manuscript:
|
Reviewer 2 Comments |
Changes incorporated by Authors
|
|
You have done a lot to make the manuscript better. Unfortunately, you did not pay attention to my remark, concerning the entire text, not only lines 21-31. One of my remarks said: The reviewer is sure that just using the digits to refer the reader to the source without mentioning the source itself (e.g., the name of the scholar) is a misleading and sometimes frustrating option: why should the reader turn the pages each time he/she meets a source? From line 21 to line 41 (13 lines) we came across six sources. Do we need to turn to the end of the article each time to see who it was? We suggest saying "Lv, C. et.al [1] states that...." looking at (reading) lines 42-51 (and further), we see sentences and references to their sources and all these look like mottoes. How are they different or close? Are they contradicting another author's view(s) or adding something? |
|

Round 3
Reviewer 1 Report
Thank you for addressing most of my comments. However, my last concerns are as following:
Theoretical framework
More explanation needed regarding the theory(s) used. What di mean by UAT. I thins this abbreviation is commonly used for uniform asymptotic theory (UAT), Is tis theory was applied as a theoretical base in your study? In fact, the presented factors revealed that you have used unified theory of acceptance and use of technology (UTAUT). Kindly double check and add more clarification with this regards. However, I didnt see any variable in the proposed model related to UTAUT factors (more justification is needed and more references that support this proposal).
Theoretical framework
Kindly consider UAT (for the whole manuscript and location) and revise accordingly if needed.
Good luck
Author Response
Manuscript ID: sustainability-2186364
The paper titled ‘Machine Learning Adoption in Educational Institutions: Role of Internet of Things and Digital Educational Platforms’ has been reviewed changes are incorporated and highlighted in the revised manuscript:
|
Reviewer 1 Comments |
Changes incorporated by Authors
|
|
Theoretical framework More explanation needed regarding the theory(s) used. What di mean by UAT. I think this abbreviation is commonly used for uniform asymptotic theory (UAT), Is tis theory was applied as a theoretical base in your study? In fact, the presented factors revealed that you have used unified theory of acceptance and use of technology (UTAUT). Kindly double check and add more clarification with this regards. However, I didnt see any variable in the proposed model related to UTAUT factors (more justification is needed and more references that support this proposal). Theoretical framework Kindly consider UAT (for the whole manuscript and location) and revise accordingly if needed.
|
Thanks These changes are incorporated by rewriting the Theoretical framework section. Add more arguments in the light of UTAUT. UAT instead of UTAUT is corrected in the whole manuscript. |

Reviewer 2 Report
Dear authors
My thanks go to you for the manuscript brushing and making it shine.
Author Response
Manuscript ID: sustainability-2186364
The paper titled ‘Machine Learning Adoption in Educational Institutions: Role of Internet of Things and Digital Educational Platforms’ has been reviewed changes are incorporated and highlighted in the revised manuscript:
|
Reviewer 2 Comments |
Changes incorporated by Authors
|
|
My thanks go to you for the manuscript brushing and making it shine. |
Thanks for accepting the changes.
|
